# The Relationship between Nursing Students’ Smart Devices Addiction and Their Perception of Artificial Intelligence

**DOI:** 10.3390/healthcare11010110

**Published:** 2022-12-30

**Authors:** Sally Mohammed Farghaly Abdelaliem, Wireen Leila Tanggawohn Dator, Chandrakala Sankarapandian

**Affiliations:** 1Department of Nursing Management and Education, College of Nursing, Princess Nourah bint Abdulrahman University, P.O. Box 84428, Riyadh 11671, Saudi Arabia; 2Department of Medical–Surgical Nursing, Princess Nourah bint Abdulrahman University, P.O. Box 84428, Riyadh 11671, Saudi Arabia

**Keywords:** nursing, students, smart devices, addiction, perception, artificial intelligence

## Abstract

Background: The concept of addiction in relation to cellphone and smartphone use is not new, with several researchers already having explored this phenomenon. Artificial intelligence has become important in the rapid development of the technology field in recent years. It has a very positive impact on our day-to-day life. Aim: To investigate the relationship between nursing students’ addiction to smart devices and their perceptions of artificial intelligence. Methods: A cross-sectional design was applied. The data were collected from 697 nursing students over three months at the College of Nursing, Princess Nourah bint Abdulrahman University. Results: The correlation test shows a significant correlation between smart device addiction and the artificial intelligence of the respondents (*p*-value < 0.05). In addition, the majority of the students, 72.7% (507), are moderately addicted to smartphones, 21.8% (152) are highly addicted, and only 5.5% (38) have a low addiction. Meanwhile, 83.6% (583) of them have high levels of perception of artificial intelligence and the rest, 16.4% (114), have a moderate level. Conclusions: The nursing students’ perception of artificial intelligence varies significantly according to their level of addiction to smart device utilization.

## 1. Introduction

Technology is regarded as one of the key important advancements of educational systems [1]. Mobile phones have become a ubiquitous part of everyday life. Nomophobia denotes the fear of an individual when they are unable to access their mobile phone, due to no signal or low battery. “Nomo” means no mobile and “Phobia” means fear. Such a fear can even disturb their concentrating ability. Nomophobia is viewed as a digital disease of the 21st century [2]. According to findings by Qutishat et al. (2020) [3], Nomophobia has a significant impact on the educational attainment of learners. The effects mentioned were a lack of class attendance, reduced studying habits and grading, poor concentration, and regular late attendance for class. The cause for the above effects result from decreased sleep during the night and excessive engagement with mobiles. Moro et al. (2021) [4] posited that while technology enhanced the learning environment, creating a positive class atmosphere, it also created the addiction to smart devices. Nomophobia implies feelings of dissidence, anxiety, and agony due to the inability to access the mobile phone [5]. The research findings revealed that the overuse of mobile technology in teenagers leads to anti-sociability, technology addiction, and negatively affects their academic performances [6,7]. When they cannot access their mobile phones, it causes them to have anxiety, depression, and agitation and nervous tension [8,9].

Addiction to cellphones and smartphone use is not a new phenomenon, as several studies have shown [10,11]. In the literature, the terms “cellphone” and “smartphone” have been used interchangeably, but smartphones are simply cellphones with advanced features, such as the ability to download and use apps and access the internet [12]. Cellphone addiction was previously defined as problematic cellphone or smartphone use [13]. Cellphone addiction is defined as a behavioral addiction, a disorder characterized by behaviorally expressed symptoms associated with a pleasurable and irresistible quality [14]. Cellphones are an important part of most college students’ daily lives, serving as a tool for social interaction, information retrieval, and entertainment. The Pew Research Center (2018) [15] reports that 99% of young adults aged 18–29 own a cellphone, with 96% owning a smartphone. The availability of smartphones allows for instant gratification, but at a cost. College students spend far too much time on their cellphones, according to one study [16], with one study estimating that they spend nearly 9 h per day on the phone. Indeed, increased phone use has been linked to lower grades, possibly due to college students using their phones in class [17]. Cellphone use also has an effect on the mental and physical health of college students. Although the researchers did not define it, excessive cellphone use is associated with poorer sleep quality, increased anxiety, and a shorter life span. The desire to determine what constitutes cellphone addiction is widespread [18].

The use of artificial intelligence (AI) is an important evolvement in education and clinical practice among nursing students. Now, smart health monitoring is performed effectively by programmable object interfaces. These technologies help to monitor the medical services such as medical nursing and rehabilitation, patient observations, evaluation, and screening system. Tele-health is the most useful invention in medical advancement [19]. The term “Artificial Intelligence” is related to measuring human-like intelligence in computers [20]. Artificial intelligence has become increasingly important in the rapidly developing field in recent years. It has a very positive impact on our daily life [21]. The Ministry of the Republic of Turkey’s National Education in 2018 carried out a curriculum evaluation of revised technology incorporated curricula. The findings showed that students were equipped with good knowledge and problem-solving techniques, were critical thinkers, adventurous and in control, had communication skills, could empathize, and were useful contributors to society and culture [22]. The most important areas of utilization of artificial intelligence in the health field are health monitoring, handling patient data, medication evolution, surgery, telemedicine, health statistics, and individualized care and the visualization of investigations [23].

Artificial intelligence-based training in medical and para-medical education will provide a chance for students to learn the use of artificial intelligence tools to solve clinical problems effectively [24]. Multiple authors (Barret et al., 2019, and Dwivedi et al., 2021) have stated that dynamic AI techniques will unseal clinically useful information concealed in the huge sum of data, this will support for officiating clinical choices [25,26]. Littman et al. (2021) [21] studied university students’ perceptions about artificial intelligence among 130 fourth-year students from departments such as education, economics, administrative science, and the Arts in the Eastern Anatolia region in Turkey. The findings reveled that, students from education had a higher perception of artificial intelligence compared to the students of the faculty of economics, administrative sciences, and the Arts department. The negative perception of all the sample groups was present regarding the artificial intelligence concept. Therefore, the researchers were recommended to present lectures to the students about the usage of artificial intelligence in education and practical fields to cross examine their negative perception toward artificial intelligence. Briganti and Le Moine (2020) [27] quoted that many researchers opinions on artificial intelligence will have an important role in medical and nursing fields in the upcoming years. He recommended to introduce AI to the medical and nursing school’s curriculum to provide insight about AI algorithms and use.

The survey of data from health care systems will cause evolutionary changes in personalized medicine in future [21]. Chang, Lai, and Hwang (2018) [28] described that integrating innovative technologies helps nursing training under little clinical tutoring. It also helps for health care practitioners to conduct training and professional skills effectively. The domain of university education has changed greatly. Internet technologies have assumed a vital place. With the emergence of mobile technologies. Now, learners and teachers construct their own resources and interconnect with one another. When these technologies are used effectively, it also helps to improve the self-learning experiences among them [29]. Many studies were performed on smartphone addiction and the perception of AI among nursing students. Based on these points, the present study aimed to identify the association between nursing students’ smart device dependency and their perception on artificial intelligence. As a result, the aim of this study is to investigate the relationship between nursing students’ smart device addiction and their perception of artificial intelligence. The study also addressed the following research questions: “What are the levels of students’ smart device addictions and their perceptions of artificial intelligence at the Princess Nourah bint Abdulrahman University in Riyadh?” and “What is the relationship between nursing students’ smart device addictions and their perceptions of artificial intelligence?” Furthermore, the study hypothesis stated that there is a significant positive relationship between nursing students’ smart device addictions and their perceptions of artificial intelligence.

## 2. Materials and Methods

### 2.1. Research Design, Setting, and Sample

A quantitative, descriptive, correlational, and cross-sectional research study design was conducted to investigate the relationship between nursing students’ smart device addictions and their perceptions of artificial intelligence at the college of Nursing in Princess Nourah bint Abdulrahman UniversityRiyadh, Saudi Arabia during the academic 2021/2022 year. The population for this study included baccalaureate female nursing students who were enrolled from first year to internship year. A total of 712 nursing students (Target Population) were recruited (whole-population sampling) from the study setting. Of them, 697 (97.9%) agreed to participate in the study and completed the self-administered questionnaire. The data collection was performed over 3 months from September to November 2021.

### 2.2. Study Instruments

Age, current academic level, and specialty in the current academic level were among the demographic data collected. Smart device addiction was assessed using the smartphone addiction tool [30]. This scale consisted of 33 items. The participants rated each item on a 6-point scale (1 = strongly disagree, 6 = strongly agree). The researchers computed the average score and the overall scale score; higher scores indicate higher levels of smartphone addiction. The concept validity, criteria validity, and internal consistency reliability of the scale were all determined to be adequate (α = 0.880).

To assess artificial intelligence, the researchers developed the artificial intelligence questionnaire based on: (a) the technology readiness index (TRI 2.0), which was developed by Parasuraman and Colby (2015) [31] to measure technology readiness, and (b) the technology acceptance model (TAM), which was developed by Davis (1989) [32] to measure technology acceptance. The questionnaire consisted of 30 items: 16 items for measuring technology readiness (i.e., 4 items for optimism, 4 items for innovativeness, 4 items for discomfort, and 4 items for insecurity); 12 items for measuring technology acceptance (i.e., 6 items for perceived usefulness and 6 items for perceived ease of use); and 2 items concerning the intention to adopt artificial intelligence technologies were designed specifically for this study (i.e., “I consider using artificial intelligence technologies as a nursing student” and “I will use artificial intelligence technologies when performing nursing practices as a professional nursing student”). Technology readiness items were measured on a 5-point Likert scale anchored by strongly disagree = 1 and strongly agree = 5. Technology acceptance items were measured on a 7-point numeric scale anchored by extremely unlikely = 1 to extremely likely = 7. The average score for the entire questionnaire score was calculated by the researchers (α = 0.764). The researchers computed the average score and the overall scale score; higher scores indicate higher level of perception regarding artificial intelligence. The concept validity, criteria validity, and internal consistency reliability of the scale were all determined to be adequate.

### 2.3. Validity and Reliability

The two tools were adjusted then translated into Arabic and back into English. The tools were then submitted to a panel of five experts (four Professors and one Lecturer from the Nursing Administration Department) who examined and assessed the content validity and offered feedback on the content, question types, and item clarity. Their comments were considered to ensure accuracy and to prevent possibly undermining the study. To examine the reliability of research tools, the internal consistency of items was measured using the Cronbach’s alpha coefficient test. At a statistical significance level of *p* ≤ 0.05, the three tools were determined to be reliable, with α = 0.880 for tool one and 0.764 for tool two. The pilot study was performed on 10% different students from different health colleges (*n* = 70) to assess item clarity and practicality, identify potential hurdles and concerns during data collection, and test the time necessary to complete the questionnaire. Some aspects need clarification from researchers but did not necessitate change. The participants in the pilot study were not included in the study sample.

### 2.4. Data Collection

The data were gathered via survey questionnaires, which were distributed online to nursing students. The data were collected for three months, from September to November 2021. The data were collected with the consent of the nursing students during the agreed break period. The time required to fill out the questionnaires was 10 min. 

### 2.5. Ethical Considerations

The Institutional Review Board of the Princess Nourah bint Abdulrahman University (N: 21-0196) excused the study from ethical assessment. The subjects provided informed consent after being told about the goal of the study. Confidentiality and anonymity were ensured by assigning a code number to each questionnaire. The students were assured that their information would be kept strictly confidential and used only for research purposes. The ability to exit the study at any time has been ensured.

### 2.6. Data Analysis

SPSS version 23 was used to analyze the collected data. To quantify demographic characteristics, descriptive statistics (frequency, means, standard deviations, and percentages) were used to measure the normality of the continuous data, whereas inferential statistics such as the Mann–Whitney test and Kruskal–Wallis test were used for abnormally distributed quantitative variables, to compare between two studied groups. To analyze the link between the variables in the study, the Pearson correlation coefficient was used. To predict the most affecting factor for artificial intelligence, a hierarchical linear (Stepwise) regression analysis was undertaken. The variables included as independent variables in the multiple regression models were those that were statistically significant (*p* -value ≤ 0.05) in the correlational analysis, with a correlation coefficient of 100.

## 3. Results

### 3.1. Demographic Characteristics of The Study Participants

Table 1 revealed that more than two thirds (64.3%) of the nursing students were within the age group ranged from 20 to 25 years old. Furthermore, the female nursing students were distributed as following among the nursing academic levels: 34.7% of the students were enrolled within the first year, 17.6% of them were enrolled within second year, 13.9% were enrolled within third year, 16.6% were enrolled within fourth year, and 17.1% of them were enrolled within the internship year. Moreover, more than one third (36.2%) of the nursing students were enrolled within the clinical specialties of Psychiatric, Community, Critical, and Nursing Management, while 14.3% of the nursing students were enrolled within the nursing specialty of Maternity and Pediatric Nursing.

### 3.2. Level of Smartphone Addiction and Artificial Intelligence

The findings (Table 2) showed that the majority of the respondents (507) 72.7% were moderately addicted to their smartphones, while (152) 21.8% were highly addicted, and only (38) 5.5% were minimally addicted. Meanwhile, (583) 83.6% of the nursing students had high levels of perception regarding artificial intelligence and the rest (114) 16.4% had moderate levels of perception regarding artificial intelligence. In terms of their artificial intelligence, the majority (75.5%) of the nursing students had moderate technology readiness, (89.4%) high levels of technology acceptance, and 65.1% of the nursing students had high levels of AI technology adoption. Regarding technology readiness, more than half of the respondents had high levels of optimism (53.5%) and insecurity (57%). Additionally, the majority (40.2%) had moderate levels of innovativeness (40%) and discomfort (60%). The perceived usefulness of technology was high (97.1%) and the perceived ease of use was also high (78.3%).

### 3.3. Correlation between Smartphone Addiction and Artificial Intelligence

Table 3 showed a highly significant correlation between smartphone addiction and artificial intelligence among nursing students (*p*-value < 0.001). The specific domains of artificial intelligence that include technology readiness, technology acceptance, and technology adoption were all negatively correlated to smartphone addiction as shown by the negative r values. Only discomfort and insecurity scales under technology readiness were positively correlated to smartphone addiction. These findings were significant with respective *p*-values less than 0.05 level of significance. Among the artificial intelligence domains, the correlation test revealed significant positive correlation among technology readiness, technology acceptance, and technology adoption with corresponding *p*-values less than the 0.001 level of significance. Specific sub-domains for technology readiness: optimism, innovativeness, discomfort, and insecurity were significantly correlated to the perceived usability and perceived ease of use scales of technology acceptance. Negative correlation was found among optimism, discomfort, and insecurity; the same applies with innovativeness to discomfort, insecurity, and perceived usability. Meanwhile, discomfort and insecurity are also negatively correlated to perceived usefulness, perceived ease of use, and technology adoption.

### 3.4. Correlation between Nursing Students’ Demographic Characteristics and the Study Variables

Table 4 showed that there was a highly significant correlation found between students’ smartphone addictions with their age, academic level, and specialty of their academic level (r = 48,811.0, *p*-value < 0.001, r 42.470, *p*-value < 0.001, and r = 36.882, *p*-value < 0.001), respectively. Additionally, a highly significant correlation was found between artificial intelligence and the nursing students’ age, academic level, and specialty of their academic level (r = 47,799.0, *p*-value < 0.001, r 29.308, *p*-value < 0.001, and r = 81.394, *p*-value < 0.001), respectively.

### 3.5. Hierarchical Linear Regression (Stepwise) of Artificial Intelligence

Table 5 reflects that the regression analysis model showed that the R2 = 0.580, which means that only 47.2% of nurses’ overall perception regarding artificial intelligence was explained by smartphone addiction; F-value = 238.916 (*p*-value < 0.001), this indicates that the model is significant. Additionally, there is a highly significant variance in the degree of the associations of overall nursing student’s smartphone addiction (independent variables) with the dependent variable. 

## 4. Discussion

This research examined at nursing students’ smartphone addiction and their perceptions of artificial intelligence. The current study discovered that smartphone addiction was linked to nursing students’ perceptions of artificial intelligence. Furthermore, artificial intelligence domains were correlated and the demographic profile was correlated to the students’ smartphone addiction as well as artificial intelligence and its specific domains. It was discovered that the majority of students (72.7%) are moderately addicted to smartphones, while 21.8% are severely addicted. The findings confirmed general observations about students’ excessive engagement with smartphones. Valsaraj et al. (2019) [33] reported similar findings among nursing students in Oman, where smartphone addiction is also prevalent. Celikkalp et al. (2020) [34] described smartphone addiction among Turkish nursing and medical students. Although the level was below average, Birgül Cerit, Nevin tak Bilgin, and Bedriye Ak (2018) [35] discovered smart addiction among nursing students. Smartphone use has undoubtedly taken center stage in this technological age, even among nursing students. The use of technology, including the use of smartphones, is clearly on the rise. Smartphone use effectively bridged the digital divide, facilitating information access, online shopping and business [36,37], socialization and connectivity beyond physical distance, entertainment [27], and even academic instruction, learning, and education [38,39]. This ease of use effectively hooked users, including students, on smartphone reliance. Unfortunately, it was a genuine and pressing issue that required attention. Smartphone addiction has also been linked to negative outcomes such as poor interpersonal communication, sleep disruption, and even interference with clinical practice in health professionals [40]. Furthermore, Obanolu et al. (2021) [41] discovered that it had a negative impact on students’ academic performance, skills, and learning. Digital addiction is on the rise, especially with the unstoppable evolution of the smartphone and all its appealing affordances.

Female nursing students’ age and academic year were positively related to smartphone addiction. Andone et al. (2016) [42] discovered that the daily mean usage of smartphones is highest among younger users and decreases with age. A more recent study, however, reported similar findings in terms of age consistent with the results of this study from the executive summary report on Deloitte’s 2017 [43] Global Mobile Consumer Survey performed for seven consecutive years, which revealed an upward trend of users in terms of age. This is due to the fact that smartphone apps not only provide entertainment but are also excellent and convenient for online business transactions, health apps, communication and information, and a variety of other apps that cause daily life activities to be more convenient not only for office workers but also for homebound and mobile adults. Although smart addiction or digital dependency was unavoidable, the same Deloitte’s (2017) [43] survey report also highlighted that smartphone users had become more conscious of digital etiquette, which includes phone use while eating, talking to people, or engaging in other interactive activities. This could imply that nurse scholars and scientists can take advantage of nursing students’ digital dependency or smartphone addiction to begin developing nursing apps for education and practice that students and nurses can easily access.

The current study’s nursing students’ artificial intelligence was overwhelmingly high, with a percentage of 83.6% having no insecurity. This finding was remarkable because many scholars considered artificial intelligence applications to be promising in facilitating patient care activities and improving nurse work processes [44]. However, the nursing students’ enthusiasm for artificial intelligence may not be fully utilized in terms of their education. According to Ronquillo et al., 2017 [45], and Topaz et al., 2016 [46], nursing education is still far from incorporating artificial intelligence into the curriculum and continues to grope in the standardization and integration of informatics competencies. The global health care system has increasingly incorporated artificial intelligence, with nursing remaining uninvolved. Nursing was far behind the artificial intelligence bandwagon, as evidenced by the scarcity of critical discourse on artificial intelligence in the nursing literature [45,46].

This study’s nursing students demonstrated a high level of technology readiness, acceptance, and adoption. The perceived usefulness and perceived ease of use of technology among female nursing students are affected by their level of discomfort and insecurity, as evidenced by the negative correlation found between these variables. This finding was also discovered by Tung et al. (2020) [47], who considered ease of use in user interface design as a predictor of technology acceptance. The perceived ease of use and usefulness were deemed important in determining whether or not to use the technology for their education. Despite their digital addiction, perceived usefulness and ease of use remained major concerns for their learning and communication.

## 5. Conclusions

This study researched the relationship between nursing students’ smart device addictions and their perceptions of artificial intelligence. According to the findings of this study, students’ smartphone addictions have a highly statistically significant correlation with their perceptions regarding artificial intelligence. Moreover, the majority of nursing students had a moderate level of smartphones addiction and the majority of them had a high perception level regarding artificial intelligence. The findings of this study in terms of the level of artificial intelligence indicated that the nursing students were equipped to embrace artificial intelligence; however, the nursing profession seems unable to match the same enthusiasm. To bridge this gap, it is imperative to develop standard artificial intelligence competencies integrated in the nursing curriculum, meaning all students and entry level nurses must receive education.

## 6. Recommendations

Based on the study findings, the following further practices are recommended to be applied in order to cover the gap between education and practice: large multi-centers and cross-cultural surveys are required to be conducted to assess and capture the nursing student’s feedback on smartphone dependence levels and knowledge regarding artificial intelligence in the health field by considering their gender, culture, and age factors; nursing student’s mental health assessments need to be compared with the levels of smart device addiction; a future study can be conducted to find out the cause for mobile phone addiction and various underlying pathologies such as anxiety disorders, impulsive control deficit, and personality factors; a long-term study can be conducted to find out the effect of Nomophobia and interventions to prevent any adverse effects; educational games and gamification techniques can be used by the nursing students to supplement the learning beyond the traditional classrooms; the nursing curriculum needs to be updated according to the changes and the influence and impact of artificial intelligence on medical and social changes in the contemporary life of people at the national and international level; and recent information in the field of artificial intelligence application and usage in the clinical and educational field need to be incorporated in the nursing curriculum.

## 7. Strengths and Limitations

The findings of this study significantly added to the existing research on smartphone addiction and artificial intelligence. The study, however, should be interpreted in light of its limitations. The participants were drawn from a specific setting, so the generalizability of the results is limited. Furthermore, because the current results were based on self-reported data, they were vulnerable to response bias and subjectivity. Additionally, this study only showed correlations between study variables; no causal relationship can be established. In the future, longitudinal, experimental, and multi-site research may help to address these limitations. The current study had several advantages; as the cross-sectional method allowed for the simultaneous measurement of multiple variables in a population sample, it resulted in more reliable data that was less susceptible to the potential biases of case series and case reports. A longer follow up could have aided the investigation. Finally, no claim was presented about the relationship between the variables in the study, as its purpose was to look into the relationship between variables. Future research should focus on specific strategies for developing standardized guidelines for artificial intelligence application among nursing students and the nursing profession. Future research can also test reasons for nursing students’ smartphone addictions, as well as the effects of smartphones addiction on the physical, psychological, and social wellbeing of the nursing students.

## Figures and Tables

**Table 1 healthcare-11-00110-t001:** Distribution of the studied nursing students according to their demographics (*n* = 697).

Demographic Data	No.	%
Age (years)		
>20	249	35.7
20–25	448	64.3
Your current academic level		
First Year	242	34.7
Second year	123	17.6
Third year	97	13.9
Fourth year	116	16.6
Internship year	119	17.1
Specialty of your current academic level		
Maternity and Pediatric	100	14.3
Psychiatric, Community, Critical, Nursing Management	252	36.2
Adult Nursing Care	173	24.8
Health Assessment and Fundamentals of Nursing	172	24.7

**Table 2 healthcare-11-00110-t002:** Mean and standard deviation of the study variables (*n* = 697).

Study Variables	Mean Score	Low (<33.3)		Moderate (33.3–66.67)		High (≥66.67)	
	Mean ± SD.	No.	%	No.	%	No.	%
Smartphone Addiction	3.21 ± 0.53	38	5.5	507	72.7	152	21.8
Artificial Intelligence							
Technology Readiness							
Optimism	3.81 ± 0.66	0	0.0	324	46.5	373	53.5
Innovativeness	3.19 ± 1.03	170	24.4	280	40.2	247	35.4
Discomfort	2.84 ± 0.72	184	26.4	418	60.0	95	13.6
Insecurity	3.85 ± 0.67	0	0.0	300	43.0	397	57.0
Overall Technology Readiness	3.18 ± 0.50	38	5.5	526	75.5	133	19.1
Technology acceptance							
Perceived Usefulness	6.36 ± 0.73	0	0.0	20	2.9	677	97.1
Perceived Ease of Use	5.70 ± 1.05	0	0.0	151	21.7	546	78.3
Overall Technology Acceptance	6.03 ± 0.76	0	0.0	74	10.6	623	89.4
Technology Adoption	5.12 ± 1.40	36	5.2	207	29.7	454	65.1
Artificial Intelligence	4.45 ± 0.53	0	0.0	114	16.4	583	83.6

**Table 3 healthcare-11-00110-t003:** Correlation matrix between the study variables (*n* = 697).

Study Variables		Smartphone Addiction	Artificial Intelligence									
			Technology Readiness					Technology Acceptance			Technology Adoption	Overall Artificial Intelligence
			Optimism	Innovativeness	Discomfort	Insecurity	Overall Technology Readiness	Perceived Usefulness	Perceived Ease of Use	Overall Technology Acceptance		
Smartphone Addiction	r		−0.382 *	−0.514 *	0.421 *	0.399 *	−0.638 *	−0.325 *	−0.415 *	−0.444 *	−0.439 *	0.654 *
	*p*		<0.001 *	<0.001 *	<0.001 *	<0.001 *	<0.001 *	<0.001 *	<0.001 *	<0.001 *	<0.001 *	<0.001 *
Technology Readiness												
Optimism	r			0.409 *	−0.551 *	−0.364 *	0.509 *	0.348 *	0.417 *	0.458 *	0.279 *	0.568 *
	*p*			<0.001 *	<0.001 *	<0.001 *	<0.001 *	<0.001 *	<0.001 *	<0.001 *	<0.001 *	<0.001 *
Innovativeness	r				−0.215 *	−0.262 *	0.763 *	−0.037	0.302 *	0.192 *	0.298 *	0.548 *
	*p*				<0.001 *	<0.001 *	<0.001 *	0.335	<0.001 *	<0.001 *	<0.001 *	<0.001 *
Discomfort	r					0.388 *	−0.640 *	−0.371 *	−0.407 *	−0.462 *	−0.274 *	−0.635 *
	*p*					<0.001 *	<0.001 *	<0.001 *	<0.001 *	<0.001 *	<0.001 *	<0.001 *
Insecurity	r						−0.672 *	−0.122 *	−0.077 *	−0.112 *	−0.080 *	−0.417 *
	*p*						<0.001 *	0.001	0.042	0.003	0.036	<0.001 *
Overall technology readiness	r							0.204 *	0.395 *	0.372 *	0.314 *	0.773 *
	*p*							<0.001 *	<0.001 *	<0.001 *	<0.001 *	<0.001 *
Technology acceptance												
Perceived Usefulness	r								0.424 *	0.777 *	0.497 *	0.635 *
	*p*								<0.001 *	<0.001 *	<0.001 *	<0.001 *
Perceived Ease of Use	r									0.899 *	0.479 *	0.798 *
	*p*									<0.001 *	<0.001 *	<0.001 *
Overall Technology Acceptance	r										0.573 *	0.861 *
	*p*										<0.001 *	<0.001 *
Technology Adoption	r											0.663 *
	*p*											<0.001 *
Overall artificial intelligence	r											
	*p*											

r: Pearson coefficient. *: Statistically significant at *p* ≤ 0.05.

**Table 4 healthcare-11-00110-t004:** Correlation between nursing students’ demographics and the study variables (*n* = 697).

Demographic Data	*N*	Smartphone Addiction	Artificial Intelligence			
			Technology Readiness	Technology Acceptance	Technology Adoption	Artificial Intelligence
		Mean ± SD.	Mean ± SD.	Mean ± SD.	Mean ± SD.	Mean ± SD.
Age (years)						
>20	249	3.28 ± 0.56	3.34 ± 0.45	6.08 ± 0.58	4.93 ± 1.51	4.54 ± 0.45
20–25	448	3.18 ± 0.51	3.10 ± 0.51	6.01 ± 0.84	5.22 ± 1.33	4.40 ± 0.56
*U*(*p*)		48,811.0 * (0.006 *)	39,802.0 * (<0.001 *)	53,734.0 (0.422)	51,670.0 (0.104)	47,799.0 * (0.002 *)
Your current academic level						
First Year	242	3.29 ± 0.57	3.28 ± 0.47	5.98 ± 0.65	4.85 ± 1.50	4.46 ± 0.49
Second year	123	3.34 ± 0.58	3.19 ± 0.68	6.14 ± 0.85	5.77 ± 1.27	4.54 ± 0.70
Third year	97	3.17 ± 0.51	3.21 ± 0.47	6.07 ± 0.86	5.13 ± 1.41	4.48 ± 0.57
Fourth year	116	3.15 ± 0.51	3.26 ± 0.42	5.84 ± 0.88	4.72 ± 1.49	4.39 ± 0.55
Internship year	119	3.02 ± 0.34	2.88 ± 0.28	6.20 ± 0.59	5.37 ± 0.85	4.38 ± 0.29
*H*(*p*)		42.470 * (<0.001 *)	72.751 * (<0.001 *)	23.695 * (<0.001 *)	58.534 * (<0.001 *)	29.308 * (<0.001 *)
Specialty of your current academic level						
Maternity and Pediatric	100	3.20 ± 0.66	3.37 ± 0.57	6.44 ± 0.52	6.31 ± 0.76	4.79 ± 0.43
Psychiatric, Community, Critical, Nursing Management	252	3.13 ± 0.45	3.15 ± 0.39	6.0 ± 0.81	4.96 ± 1.29	4.41 ± 0.47
Adult Nursing Care	173	3.41 ± 0.44	2.86 ± 0.40	5.81 ± 0.80	5.21 ± 1.12	4.20 ± 0.53
Health Assessment and Fundamentals of Nursing	172	3.16 ± 0.60	3.46 ± 0.49	6.07 ± 0.65	4.56 ± 1.67	4.58 ± 0.52
*H*(*p*)		36.882 * (<0.001 *)	117.811 * (<0.001 *)	48.631 * (<0.001 *)	117.997 * (<0.001 *)	81.394 * (<0.001 *)

SD: Standard deviation; *U*: Mann–Whitney test; *H*: H for Kruskal–Wallis test. *p*: *p* value for comparing between the studied categories. *: Statistically significant at *p* ≤ 0.05.

**Table 5 healthcare-11-00110-t005:** Hierarchical linear regression (stepwise) of artificial intelligence.

Study Variables	B	Beta	*t*	*p*	95% CI	
					LL	UL
Smartphone addiction scale	0.472	0.703	28.222 *	<0.001 *	0.505	0.439
R^2^ = 0.580, F = 238.916 *, *p* < 0.001 *						

F, *p*: f and *p* values for the model. R^2^: Coefficient of determination; B: Unstandardized Coefficients Beta: Standardized Coefficients. *t*: *t*-test of significance; OR: Odds ratio; CI: Confidence interval; LL: Lower limit; UL: Upper Limit; *: Statistically significant at *p* ≤ 0.05.

## Data Availability

The datasets generated and/or analyzed during the current study are not publicly available due to data privacy but are available from the corresponding author on reasonable request.

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
