# Peer review of "The Relationship between Nursing Students’ Smart Devices Addiction and Their Perception of Artificial Intelligence"

_healthcare, 2022, doi:10.3390/healthcare11010110_

Round 1
Reviewer 1 Report
1. The paper presents an interesting and topical research paper on relationship between nursing students' smart devices addiction and their perception of artificial intelligence. From a formal point of view, I recommend checking the article (small and capital letters in title).
2. To improve the paper quality, please write more structural on discussion to identify what is the usefulness for nursing student based on statistical report, than re-writing the result.
Author Response
Dear Respected Professor Reviewer,
After warm greetings,
We would like to thank you for your constructive review and attached the responses to your valuable recommendations for research improvement:
- The paper presents an interesting and topical research paper on relationship between nursing students' smart devices addiction and their perception of artificial intelligence. From a formal point of view, I recommend checking the article (small and capital letters in title).
Authors response: The required modifications in the title was done as per your valuable recommendation
- To improve the paper quality, please write more structural on discussion to identify what is the usefulness for nursing student based on statistical report, than re-writing the result.
Authors response: Rewriting with focus structural way of writing was done as per your valuable recommendations to improve the quality of the research.

Reviewer 2 Report
Dear authors,
This is an interesting topic for the scientific community and society at large on the addiction to smart devices and the perception of artificial intelligence.
However, I propose some improvements:
- Abstract: the aim of the research should come after the background, not before.
- The title is appropriate and understandable.
- The keyword "Relationship" is not appropriate. It does not provide information about the study carried out. It could be deleted.
- Introduction: there are enough previous studies that are very up to date. In general, this section is good and provides good information.
The numbers of previous studies should be enclosed in square brackets [], not in brackets () or smaller.
- Participants: information about participants should be in section 2, not in section 3 "Demographic characteristics of participants". I propose moving the data from one section to the other.
- I suggest adding authors who support the method used (line 122).
- Instrument: internal consistency values are provided.
- The Institutional Review Board of Princess Nourah bint Abdulrahman University (N: 178 21-0196) excused the study from ethical assessment.
- The data analysis section is very well developed and responds to the essential analyses in quantitative research, but was the normality or non-normality of the data analysed?
- Results: the letters n, N, SD, U, H, t and p should be in italics (these are found in the tables).
- This study showed a highly significant correlation between smartphone addiction and the artificial intelligence among nursing students. It also showed that there was a highly significant correlation were found between students' smartphone addiction with their age, academic level, and specialty of their academic level.
- Discussion: This study explored the smartphone addiction level and the level of perception regarding artificial intelligence among nursing students. The data are related to sufficient previous studies belonging to the last 5 years, which gives value to this research.
- Future prospects and limitations are presented.
- The majority of references do not conform to MDPI standards.
Author Response
Dear Respected Reviewer,
After warm greetings,
we would like to thank you for your valuable and constructive review and attached is the responses to the required modifications:
This is an interesting topic for the scientific community and society at large on the addiction to smart devices and the perception of artificial intelligence.
Authors response: We are thankful for your motivating contribution on our research paper.
However, I propose some improvements:
- Abstract: the aim of the research should come after the background, not before.
Authors response: All the recommended modifications were done.
- The title is appropriate and understandable.
Authors response: We are thankful for your motivating contribution and valuable review
- The keyword "Relationship" is not appropriate. It does not provide information about the study carried out. It could be deleted.
Authors response: all the recommended modifications were done.
- Introduction: there are enough previous studies that are very up to date. In general, this section is good and provides good information.
Authors response: We are thankful for your motivating contribution and valuable review
The numbers of previous studies should be enclosed in square brackets [], not in brackets () or smaller.
Authors response: all the recommended modifications were done.
- Participants: information about participants should be in section 2, not in section 3 "Demographic characteristics of participants". I propose moving the data from one section to the other.
Authors response: thanks for your valuable review but we need to clarify that this demographic is the data presentation of the table 1 that include the distributional frequency of the participants’ demographics. So we didn’t transfer it from section 3 to section 2.
- I suggest adding authors who support the method used (line 122).
Authors response: all the recommended modifications were done.
- Instrument: internal consistency values are provided.
Authors response: We are thankful for your motivating contribution and valuable review
- The Institutional Review Board of Princess Nourah bint Abdulrahman University (N: 178 21-0196) excused the study from ethical assessment.
Authors response: We are thankful for your motivating contribution and valuable review
- The data analysis section is very well developed and responds to the essential analyses in quantitative research, but was the normality or non-normality of the data analysed?
Authors response: We are thankful for your motivating contribution, Yes through measuring the mean and standard deviation of the study variables to test the normality of the data.
- Results: the letters n, N, SD, U, H, t and p should be in italics (these are found in the tables).
Authors response: all the recommended modifications were done.
- This study showed a highly significant correlation between smartphone addiction and the artificial intelligence among nursing students. It also showed that there was a highly significant correlation were found between students' smartphone addiction with their age, academic level, and specialty of their academic level.
Authors response: We are thankful for your motivating contribution and valuable review
- Discussion: This study explored the smartphone addiction level and the level of perception regarding artificial intelligence among nursing students. The data are related to sufficient previous studies belonging to the last 5 years, which gives value to this research.
Authors response: We are thankful for your motivating contribution and valuable review
- Future prospects and limitations are presented.
Authors response: We are thankful for your motivating contribution and valuable review
- The majority of references do not conform to MDPI standards.
Authors' response: all the recommended modifications were done, as all references were reviewed and fixed as per the journal guidelines.

Reviewer 3 Report
I read with great interest the Manuscript titled “The relationship between nursing students’ smart devices addiction and their perception of artificial intelligence”
In my opinion, the topic is interesting enough to attract the readers’ attention.
Authors should consider the following recommendations
The Manuscript should be further revised by a native English speaker to improve its readability.
Sample size calculation seems missing. It should have been done based on your primary hypothesis/aim. Please, add it if it was done. If it was not, clarify that in the methods and report it as a study limitation.
Author Response
Dear Respected Professor Reviewer,
After warm greetings,
kindly we would like to thank you for your constructive review and attached are the responses to the recommended modifications as following:
I read with great interest the Manuscript titled “The relationship between nursing students’ smart devices addiction and their perception of artificial intelligence”
Authors response: We are thankful for your motivating contribution and valuable review
In my opinion, the topic is interesting enough to attract the readers’ attention.
Authors response: We are thankful for your motivating contribution and valuable review
Authors should consider the following recommendations
The Manuscript should be further revised by a native English speaker to improve its readability.
Authors response: all the recommended modifications were done, as the manuscript was reviewed for English editing and all the required modifications were done.
Sample size calculation seems missing. It should have been done based on your primary hypothesis/aim. Please, add it if it was done. If it was not, clarify that in the methods and report it as a study limitation.
Authors response: We are thankful for your valuable review and we need to clarify that the sample size calculation was not mentioned as we targeted the whole population to be included in the study as explained in the section 2
